# Weisfeiler and Leman Go Walking: Random Walk Kernels Revisited

**Nils M. Kriege**

Faculty of Computer Science, University of Vienna, Währinger Straße 29, 1090 Vienna, Austria
Research Network Data Science @ Uni Vienna, Kolingasse 14–16, 1090 Vienna, Austria
nils.kriege@univie.ac.at

## Abstract

Random walk kernels have been introduced in seminal work on graph learning and were later largely superseded by kernels based on the Weisfeiler-Leman test for graph isomorphism. We give a unified view on both classes of graph kernels. We study walk-based node refinement methods and formally relate them to several widely-used techniques, including Morgan's algorithm for molecule canonization and the Weisfeiler-Leman test. We define corresponding walk-based kernels on nodes that allow fine-grained parameterized neighborhood comparison, reach Weisfeiler-Leman expressiveness, and are computed using the kernel trick. From this we show that classical random walk kernels with only minor modifications regarding definition and computation are as expressive as the widely-used Weisfeiler-Leman subtree kernel but support non-strict neighborhood comparison. We verify experimentally that walk-based kernels reach or even surpass the accuracy of Weisfeiler-Leman kernels in real-world classification tasks.

## 1   Introduction

Machine learning with graph-structured data has various applications, from bioinformatics to social network analysis to drug discovery and has become an established research field. Graph kernels [26, 5] and graph neural networks (GNNs) [44] are two widely-used techniques for learning with graphs, the latter of which has recently received significant research interest. Technically, various methods of both categories exploit the link between graph data and linear algebra by representing graphs by their (normalized) adjacency matrix. Such methods are often defined or can be interpreted in terms of walks. On the other hand, the Weisfeiler-Leman heuristic for graph isomorphism testing has attracted great interest in machine learning [33, 34]. This classical graph algorithm has been studied extensively in structural graph theory and logic, and its expressive power, i.e., its ability to distinguish non-isomorphic graphs, is well understood [3]. The Weisfeiler-Leman method turned out to be suitable to derive powerful and efficient graph kernels [40]. Moreover, it is closely related to GNNs [45, 31, 14], allowing to establish links, e.g., to results from logic [15].

Several results have paved the way for these insights. The seminal work by Kersting et al. [22] links algebraic methods and the combinatorial Weisfeiler-Leman algorithm showing that it can be understood in terms of walks and simulated by iterated matrix products. A comprehensive view of the expressive power of linear algebra on graphs was recently given by Geerts [13]. GNNs are closely related to algebraic methods but involve activation functions and learnable weights. Morris et al. [31] has shown that (i) the expressive power of GNNs is limited by the Weisfeiler-Leman algorithm, and (ii) that GNN architectures exist that reach this expressive power for suitable weights. Independently, Xu et al. [45] obtained the same result by using injective set functions computable by multilayer perceptrons [46]. Most recently, Geerts et al. [14] has proven that already the early GCN layer-based architectures [23] achieve the expressive power of the Weisfeiler-Leman test with only a minor modification already mentioned in the original publication.

36th Conference on Neural Information Processing Systems (NeurIPS 2022).

Random walk kernels [12, 21] have been the starting point of a long line of research in graph kernels, see [26, 5] for details. An extension of random walk kernels is based on so-called tree patterns, which may contain repeated nodes just like walks [37, 28]. A cornerstone in the development of graph kernels is the Weisfeiler-Leman graph kernel [40], which in contrast to tree pattern kernels, relies on entire node neighborhoods instead of all subsets. This restriction allowed manageable feature vectors and led to a significant improvement in terms of both running time and classification accuracy. Several other kernels based on the Weisfeiler-Leman test have been proposed, e.g., combining it with optimal assignments [24] or the Wasserstein distance [42]. Graph kernels inspired the development of neural modules based on random walks and Weisfeiler-Leman labels [27]. Chen et al. [7] introduced multilayer-kernels conceptually similar to GNN layers and relates random walk and Weisfeiler-Leman kernels in this framework for graphs, where the node labels induce a unique bijection between neighbors. Dell et al. [8] recently established the equivalence in expressive power between tree patterns formalized as tree homomorphism counts and the Weisfeiler-Leman method. On this basis, homomorphism counts were quickly adapted for graph classification [35]. The classical Weisfeiler-Leman kernel [40] as well as the works [27, 7, 35] enumerate graph features and generate (approximate) feature vectors. In contrast, the classical random walk kernel is computed using the kernel trick and thus operates implicitly in a high dimensional features space. The advantages and disadvantages of fixed-size graph embeddings are subject to recent research [25, 2]. Nowadays, random walk kernels are widely abandoned, while Weisfeiler-Leman kernels remain an important baseline method performing competitively on many real-world datasets [26, 5].

**Our contribution.** We formally relate random walks in labeled graphs and the Weisfeiler-Leman method and link random walk kernels and the Weisfeiler-Leman subtree kernel. Starting from a combinatorial perspective, we consider walk-based label refinement and relate it to the Weisfeiler-Leman method. We extend the concept to fine-grained pairwise node similarities that satisfy the kernel properties and are computed via algebraic techniques using the kernel trick similarly to random walk kernels. From the node similarities, we derive a *node-centric* walk kernel on graphs. This kernel generalizes the classical random walk kernel [12] and allows grouping of walks by their start node. We show that grouping significantly improves the expressive power of random walk kernels and, with a minor modification, reaches the expressive power of the Weisfeiler-Leman method. Kernel parameters control the grouping of walks and allow to interpolate between random walk and Weisfeiler-Leman type kernels. We verify our theoretical results on real-world graphs for which our approach reaches high accuracies, surpassing the Weisfeiler-Leman subtree kernel in some cases.

## 2 Fundamentals

We aim at establishing formal links between random walks, Weisfeiler-Leman refinement, and corresponding kernels. We review the basics of these methods and refer the reader to recent surveys for further details [26, 34]. We proceed by introducing the notation used.

### 2.1 Definitions and notation

An (undirected) *graph* $G$ is a pair $(V, E)$ with *nodes* $V$ and *edges* $E \subseteq V^2$, where $(u, v) \in E \Leftrightarrow (v, u) \in E$. A graph may be endowed with a (node) *labeling* $\sigma \colon V \to \Sigma$. The *labels* $\Sigma$ can be arbitrary structures such as (multi)sets but are typically represented by or mapped to integers. We denote sets by $\{\cdot\}$ and multisets allowing multiple instances of the same element by $\{\!\!\{\cdot\}\!\!\}$. We refer to the *neighbors* of a node $u$ by $\mathcal{N}(u) = \{v \in V \mid (u, v) \in E\}$. Two graphs $G = (V, E)$ and $H = (V', E')$ are *isomorphic*, written $G \simeq H$, if there is a bijection $\psi \colon V \to V'$ such that $(u, v) \in E \Leftrightarrow (\psi(u), \psi(v)) \in E'$ for all $u, v$ in $V$. For labeled graphs, additionally $\sigma(v) = \sigma(\psi(v))$ must hold for all $v \in V$; for two graphs with roots $r \in V$ and $r' \in V'$, $\psi(r) = r'$ must be satisfied. The map $\psi$ is called *isomorphism*. An isomorphism of a graph to itself is called *automorphism*. A labeling $\sigma$ is said to *refine* a labeling $\eta$, denoted by $\sigma \sqsubseteq \eta$, if for all nodes $u$ and $v$, $\sigma(u) = \sigma(v)$ implies $\eta(u) = \eta(v)$; we write $\sigma \equiv \eta$ if $\sigma \sqsubseteq \eta$ and $\eta \sqsubseteq \sigma$. A *label refinement* is a sequence of labelings $(\sigma^{(0)}, \sigma^{(1)}, \dots)$ such that $\sigma^{(i)}$ refines $\sigma^{(i-1)}$ for all $i > 0$. A function $k \colon \mathcal{X} \times \mathcal{X} \to \mathbb{R}$ is a *kernel* on $\mathcal{X}$, if there is a Hilbert space $(\mathcal{H}, \langle \cdot, \cdot \rangle)$ and a map $\phi \colon \mathcal{X} \to \mathcal{H}$, such that $k(x, y) = \langle \phi(x), \phi(y) \rangle$ for all $x, y \in \mathcal{X}$. A kernel on the set of graphs is a *graph kernel*. We denote by $k_\delta$ the Dirac kernel with $k_\delta(x, y) = 1$ if $x = y$ and 0 otherwise. We write vectors and matrices in bold, using capital letters for the latter, and denote the column vector of ones by $\mathbf{1}$.

## 2.2 Random walk kernels

The classical random walk kernel proposed by Gärtner et al. [12] compares two graphs with discrete labels by counting their *common walks*, i.e., the pairs of walks with the same label sequence. Let $\mathcal{W}_i(G)$ be the set of all walks of length $i$ in a graph $G$. For a walk $w = (v_1, v_2, \dots)$ let $\sigma(w) = (\sigma(v_1), \sigma(v_2), \dots)$ denote its label sequence.

**Definition 2.1** ($\ell$-step random walk kernel)**.** *Let $\lambda_i \in \mathbb{R}_{\geq 0}$ for $i \in \{0, \dots, \ell\}$ be a sequence of weights, the $\ell$-step random walk kernel is*

$$K_\times^\ell(G, H) = \sum_{i=0}^{\ell} \lambda_i \sum_{w \in \mathcal{W}_i(G)} \sum_{w' \in \mathcal{W}_i(H)} k_\delta(\sigma(w), \sigma(w')).$$

Random walk kernels can be computed based on a product graph using the kernel trick.

**Definition 2.2** (Direct product graph)**.** *For two labeled graphs $G = (V, E)$ and $H = (V', E')$ the direct product graph $G \times H$ is the graph $(\mathcal{V}, \mathcal{E})$ with $\mathcal{V} = \{(v, v') \in V \times V' \mid \sigma(v) = \sigma(v')\}$ and $\mathcal{E} = \{((u, u'), (v, v')) \in \mathcal{V}^2 \mid (u, v) \in E \wedge (u', v') \in E'\}$.*

**Lemma 2.3.** *There is a bijection between $\mathcal{W}_i(G \times H)$ and $\{(w, w') \in \mathcal{W}_i(G) \times \mathcal{W}_i(H) \mid \sigma(w) = \sigma(w')\}$, $\forall i \geq 0$.*

Using Lemma 2.3, the $\ell$-step random walk kernel is computed from the adjacency matrix $\boldsymbol{A}_\times$ of $G \times H$ by matrix products or iterated matrix-vector multiplication as

$$K_\times^\ell(G, H) = \sum_{i,j}^{|\mathcal{V}|} \left[ \sum_{k=0}^{\ell} \lambda_k \boldsymbol{A}_\times^k \right]_{ij} = \sum_{i=1}^{|\mathcal{V}|} \left[ \sum_{k=0}^{\ell} \lambda_k \boldsymbol{w}^{(k)} \right]_i$$

with $\boldsymbol{w}^{(k)} = \boldsymbol{A}_\times \boldsymbol{w}^{(k-1)}$ and $\boldsymbol{w}^{(0)} = \boldsymbol{1}$. The *random walk kernel* $K_\times$ is defined as the limit $K_\times^\infty$, where $\lambda = (\lambda_0, \lambda_1, \dots)$ is chosen such that the sum converges [12, 41]. A well-studied instantiation is the *geometric random walk kernel* using weights $\lambda_i = \gamma^i$, $i \in \mathbb{N}$, where $\gamma < \frac{1}{a}$ with $a \geq \min\{\Delta^-, \Delta^+\}$ and $\Delta^{+/-}$ the maximum in- and outdegree of $G \times H$ guarantees convergence [12]. For this choice, the kernel can be computed in polynomial-time applying an analytical expression based on matrix inversion. Efficient methods of computation have been studied extensively [43, 20]. However, theoretical and empirical results suggest that the benefit of using walks of infinite length as in the geometric random walk kernel is limited [41, 25]. Random walk kernels can be extended to score the similarity of walks instead of requiring their labels to match exactly, making them suitable for graphs with arbitrary node and edge attributes compared by dedicated kernels [17, 43, 25]. The accuracy and running time of random walk kernels can be improved by refining vertex labels in a preprocessing step leading to sparse product graphs [29, 25, 19].

## 2.3 Weisfeiler-Leman refinement and graph kernels

The Weisfeiler-Leman method, often referred to as *color refinement*, initially assigns labels $\mathsf{wl}^{(0)}(v) = \sigma(v)$ to all nodes $v$ (or uniform labels if graphs are unlabeled) and then iteratively computes new labels for all $v$ in $V$ as

$$\mathsf{wl}^{(i+1)}(v) = \left( \mathsf{wl}^{(i)}(v), \{\!\!\{ \mathsf{wl}^{(i)}(w) \mid w \in \mathcal{N}(v) \}\!\!\} \right).$$

Convergence is reached when $\mathsf{wl}^{(i)}(u) = \mathsf{wl}^{(i)}(v) \iff \mathsf{wl}^{(i+1)}(u) = \mathsf{wl}^{(i+1)}(v)$ holds for all $u, v \in V$ and we denote the corresponding stable labeling by $\mathsf{wl}^{(\infty)}$. In practice, nested multisets are avoided by applying an injective mapping to integers after every iteration. For all $j \geq i$, $\mathsf{wl}^{(j)} \sqsubseteq \mathsf{wl}^{(i)}$ holds. The stable partition $\mathsf{wl}^{(\infty)}$ is refined by the orbit labeling assigning two nodes $u, v$ the same label if and only if there is an automorphism $\psi$ with $\psi(u) = \psi(v)$.

In graph isomorphism testing, only the final stable labeling $\mathsf{wl}^{(\infty)}$ is of interest, while in graph kernels, only the first few iterations are used. The *Weisfeiler-Leman subtree kernel* [40] $K_{\mathrm{WL}}^\ell$ computes the first $\ell$ iterations and maps graphs to feature vectors counting the number of nodes in the graph for every label, which is equal to

$$K_{\mathrm{WL}}^\ell(G, H) = \sum_{i=0}^{\ell} \sum_{u \in V(G)} \sum_{v \in V(H)} k_\delta(\mathsf{wl}^{(i)}(u), \mathsf{wl}^{(i)}(v)).$$

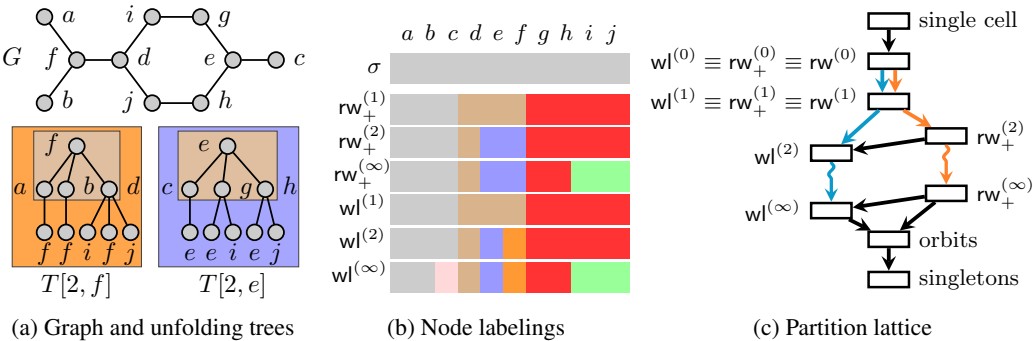

| | $a\ b\ c\ d\ e\ f\ g\ h\ i\ j$ | |
|---|---|---|
| $\sigma$ | | $\mathsf{wl}^{(0)} \equiv \mathsf{rw}_+^{(0)} \equiv \mathsf{rw}^{(0)}$ |
| $\mathsf{rw}_+^{(1)}$ | | $\mathsf{wl}^{(1)} \equiv \mathsf{rw}_+^{(1)} \equiv \mathsf{rw}^{(1)}$ |
| $\mathsf{rw}_+^{(2)}$ | | |
| $\mathsf{rw}_+^{(\infty)}$ | | |
| $\mathsf{wl}^{(1)}$ | | |
| $\mathsf{wl}^{(2)}$ | | |
| $\mathsf{wl}^{(\infty)}$ | | |

(a) Graph and unfolding trees     (b) Node labelings     (c) Partition lattice

Figure 1: Label refinement using walks and the Weisfeiler-Leman method. (a) shows an unlabeled graph $G$ and two unfolding trees of depth two rooted at node $f$ and $e$, respectively. Since the trees are not isomorphic, $\mathsf{wl}^{(2)}(f) \neq \mathsf{wl}^{(2)}(e)$. However, $\mathsf{rw}_+^{(2)}(f) = \mathsf{rw}_+^{(2)}(e)$, since the trees have the same number of nodes with uniform labels on each level. (b) shows the corresponding node partitions after one and two steps and at convergence (labels are represented by colors and reused in each iterations for simplicity) illustrating that $\mathsf{wl}^{(\infty)} \not\equiv \mathsf{rw}_+^{(\infty)}$ holds for $G$. (c) shows a part of the lattice associated with label refinement relating both methods, where an arrow from $\eta$ to $\sigma$ denotes $\sigma \sqsubseteq \eta$.

## 3 Comparing nodes by walks

We propose to compare the nodes of one or multiple graphs according to the walks originating at them. We start with discrete walk labelings and characterize their ability to distinguish nodes relating the approach to classical methods and results for unlabeled graphs, namely Morgan's algorithm, walk partitions and Weisfeiler-Leman refinement. Then we focus on the pairwise node comparison and propose walk kernels on nodes, which are extended to reach the expressiveness of the Weisfeiler-Leman method.

### 3.1 Walk labelings and their properties

We consider graphs with discrete labels. For a node $v$ in a graph $G$, we define $\mathcal{W}_i(v)$ as the set of walks of length $i$ in $G$ starting in $v$. We associate with every node $v$ a label $\mathsf{rw}^{(i)}(v)$ representing the label sequences of length $i$ walks originating at $v$, i.e., $\mathsf{rw}^{(i)}(v) = \{\!\!\{\sigma(w) \mid w \in \mathcal{W}_i(v)\}\!\!\}$. This does not yield a label refinement, since $\mathsf{rw}^{(i+1)}$ not necessarily refines $\mathsf{rw}^{(i)}$. For a counterexample consider nodes $f$ and $g$ of the graph $G$ in Figure 1(a), for which $\mathsf{rw}^{(1)}(f) \neq \mathsf{rw}^{(1)}(g)$, but $\mathsf{rw}^{(2)}(f) = \mathsf{rw}^{(2)}(g)$. To guarantee that two nodes once labeled different will obtain different labels in all subsequent iterations, we consider all walks up to a given length $\ell$ and define $\mathcal{W}_\ell^+(v) = \bigcup_{i=0}^{\ell} \mathcal{W}_i(v)$.

**Definition 3.1** ($\ell$-walk label). *The $\ell$-walk label of a node $v$ is* $\mathsf{rw}_+^{(\ell)}(v) = \{\!\!\{\sigma(w) \mid w \in \mathcal{W}_\ell^+(v)\}\!\!\}$.

Since $\mathsf{rw}_+^{(i)}(v) \subseteq \mathsf{rw}_+^{(i+1)}(v)$ for all $v$ in $V$ and walks of different length have different label sequences, it immediately follows that $\mathsf{rw}_+^{(i+1)} \sqsubseteq \mathsf{rw}_+^{(i)}$. We say that two nodes $u$ and $v$ are $\ell$-*walk indistinguishable* if $\mathsf{rw}_+^{(\ell)}(u) = \mathsf{rw}_+^{(\ell)}(v)$. We call $u$ and $v$ *walk indistinguishable* if they are $\infty$-walk indistinguishable.

#### 3.1.1 Relation to Morgan's algorithm

Morgan [30] proposed a method to generate canonical representations for molecular graphs. To reduce ambiguities each node $v$ is endowed with its *extended connectivity* $\mathsf{ec}(v)$. Let $G = (V, E)$ be a (molecular) graph, initially we assign $\mathsf{ec}^{(1)}(v) = \deg(v)$ to every node $v$ in $V$, where $\deg(v)$ is the degree of $v$. Then, the values $\mathsf{ec}^{(i)}(v)$ are computed iteratively for $i \geq 2$ and all nodes $v$ in $V$ as

$$\mathsf{ec}^{(i)}(v) = \sum_{u \in \mathcal{N}(v)} \mathsf{ec}^{(i-1)}(u),$$

until $|\{\mathsf{ec}^{(i)}(v) \mid v \in V\}| \leq |\{\mathsf{ec}^{(i+1)}(v) \mid v \in V\}|$. Then, $\mathsf{ec}^{(i)}(v)$ is the final extended connectivity of the node $v$. For all $i \geq 1$ the extended connectivity values $\mathsf{ec}^{(i)}$ are equivalent to the row (or column)

sums of the $i$th power of the adjacency matrix, i.e., $\boldsymbol{A}^i\boldsymbol{1}$, which gives the number of walks of length $i$ starting at each node [38, 10]. We relate the extended connectivity to walk labelings.

**Proposition 3.2.** *For $i \geq 1$ and all graphs, $\mathsf{rw}^{(i)} \sqsubseteq \mathsf{ec}^{(i)}$ holds with $\mathsf{rw}^{(i)} \equiv \mathsf{ec}^{(i)}$ in the case of unlabeled graphs.*

*Proof.* From the definition of walk labelings we conclude $|\mathsf{rw}^{(i)}(u)| = \mathsf{ec}^{(i)}(u)$ for all $u$ in $V$. Hence, $\mathsf{rw}^{(i)}(u) = \mathsf{rw}^{(i)}(v) \Longrightarrow \mathsf{ec}^{(i)}(u) = \mathsf{ec}^{(i)}(v)$. Vice versa, $\mathsf{ec}^{(i)}(u) = \mathsf{ec}^{(i)}(v)$ is a necessary condition for $\mathsf{rw}^{(i)}(u) = \mathsf{rw}^{(i)}(v)$. For unlabeled graphs, all walks have the same label sequence and $\mathsf{rw}^{(i)}$ contains multiple instance of a single label sequence for all nodes, proving the equivalence. $\qquad\square$

### 3.1.2 Relation to walk partitions

Powers and Sulaiman [36] studied the *walk partition* of a graph with adjacency matrix $\boldsymbol{A}$, which is obtained from the $n \times \ell$ matrix $\boldsymbol{W}^{(\ell)} = [\boldsymbol{1}, \boldsymbol{A}\boldsymbol{1}, \boldsymbol{A}^2\boldsymbol{1}, \dots, \boldsymbol{A}^\ell\boldsymbol{1}]$. We define a node labeling wp, where the $i$th node is assigned its row vector, i.e., $\mathsf{wp}(i) = \boldsymbol{W}^{(\ell)}_{i,\cdot}$. With Proposition 3.2 we directly obtain the following result.

**Proposition 3.3.** *For $i \geq 0$ and all graphs, $\mathsf{rw}^{(i)}_+ \sqsubseteq \mathsf{wp}^{(i)}$ holds with $\mathsf{rw}^{(i)}_+ \equiv \mathsf{wp}^{(i)}$ in the case of unlabeled graphs.*

For unlabeled graphs, it is known that $\mathsf{wl}^{(\infty)} \sqsubseteq \mathsf{wp}^{(\infty)}$ and both coincide for many graphs [36].

### 3.1.3 Relation to the Weisfeiler-Leman method

The Weisfeiler-Leman labels encode neighborhood patterns. The *unfolding tree* $T[n, v]$ of depth $n$ at the node $v$ is defined recursively as the tree with root $v$ and children $\mathcal{N}(v)$. Each child $w \in \mathcal{N}(v)$ is the root of the unfolding tree $T[n-1, w]$ and $T[0, w] = (\{w\}, \emptyset)$. The labels of the original graph are preserved in the unfolding tree.

**Lemma 3.4** (Folklore). *For $\ell \geq 0$ and nodes $u$ and $v$, $\mathsf{wl}^{(\ell)}(u) = \mathsf{wl}^{(\ell)}(v) \Longleftrightarrow T[\ell, u] \simeq T[\ell, v]$.*

As unfolding trees also encode walks we can relate Weisfeiler-Leman and walk labelings. For an unfolding tree $T[\ell, v]$, let $\mathsf{PL}(T[\ell, v])$ denote the set of unique paths from the root $v$ to a leaf.

**Lemma 3.5.** *Let $\ell \geq 0$ and $v$ a node, then $\mathsf{PL}(T[\ell, v]) = \mathcal{W}_\ell(v)$.*

*Proof.* For a walk $(v = v_0, \dots, v_\ell)$ we have $v_{i+1} \in \mathcal{N}(v_i)$ for all $0 \leq i < \ell$ and hence a path in $\mathsf{PL}(T[\ell, v])$ exists. Vice versa, every path in $\mathsf{PL}(T[\ell, v])$ is a walk of length $\ell$ starting at $v$. $\qquad\square$

Since $T[\ell, u] \simeq T[\ell, v]$ implies $\mathsf{PL}(T[\ell, u]) = \mathsf{PL}(T[\ell, v])$ for all $i \in \{0, \dots, \ell\}$ we conclude.

**Proposition 3.6.** *Let $\ell \in \mathbb{N}_0$, then $\mathsf{wl}^{(\ell)} \sqsubseteq \mathsf{rw}^{(\ell)}_+ \sqsubseteq \mathsf{rw}^{(\ell)}$.*

Vice versa, for $\ell = 0$ or $\ell = 1$, $\mathsf{PL}(T[\ell, u]) = \mathsf{PL}(T[\ell, v])$ implies $T[\ell, u] \simeq T[\ell, v]$, since the unfolding tree consists of a single node or is a star graph, respectively.

**Proposition 3.7.** *It holds $\mathsf{wl}^{(0)} \equiv \mathsf{rw}^{(0)}_+ \equiv \mathsf{rw}^{(0)}$ and $\mathsf{wl}^{(1)} \equiv \mathsf{rw}^{(1)}_+ \equiv \mathsf{rw}^{(1)}$.*

Figure 1 illustrates the relations and shows an example, where the refinement relation is strict. The nodes $e$ and $f$ are walk indistinguishable but obtain different Weisfeiler-Leman labels after only two refinement steps. In this sense the walk labeling is weaker than the Weisfeiler-Leman labeling. This does not necessarily mean that it is less suitable for learning tasks. We study the difference of the two techniques on common benchmark datasets in Section 5.

## 3.2 Pairwise node comparison

In the previous section, two walks were considered equal if they exhibit the same label sequence and two nodes have the same walk label if the multisets of walks originating at them are equal. We define the functions $k_\ell$ and $k_\ell^+$ between nodes that measure the similarity based on walks in a more fine-grained manner as

$$k_\ell^+(u, v) = \sum_{i=0}^{\ell} k_i(u, v) \quad \text{with} \quad k_\ell(u, v) = \sum_{w \in \mathcal{W}_\ell(u)} \sum_{w' \in \mathcal{W}_\ell(v)} \prod_{i=0}^{\ell} k_V(u_i, v_i), \tag{1}$$

where $w = (u{=}u_0, u_1, \ldots, u_\ell)$, $w' = (v{=}v_0, v_1, \ldots, v_\ell)$ and $k_V$ is a user-defined function measuring the similarity of nodes, e.g., by taking continuous attributes into account. Assuming that $k_V$ is a kernel on nodes, it follows from the concept of convolution kernels and basic closure properties [39] that $k_\ell^+$ again is a kernel. We denote its feature map by $\phi_\ell^+$ and refer the reader to [25] for the tools to construct feature vectors. If we assume that $k_V(u, v) = k_\delta(\sigma(u), \sigma(v))$, then a possible feature map $\phi_\ell^+$ takes $v$ to a vector having a component for every possible label sequence $s$ of length at most $\ell$ and counts the number of walks $w$ starting at $v$ with $\sigma(w) = s$, i.e., it is the characteristic vector of the multiset $\mathrm{rw}_+^{(\ell)}(v)$. We obtain a new parameterized kernel that allows controlling the strictness of neighborhood comparison.

**Definition 3.8** (Generalized $\ell$-walk node kernel). *Let* $\alpha \in \mathbb{R}_{\geq 0}$, *the* generalized $\ell$-walk node kernel *is*

$$\hat{k}_\ell^+(u, v; \alpha) = \exp\left(-\alpha \left\| \phi_\ell^+(u) - \phi_\ell^+(v) \right\|_2^2\right) \tag{2}$$

$$= \exp\left(-\alpha \left( k_\ell^+(u, u) + k_\ell^+(v, v) - 2k_\ell^+(u, v) \right)\right). \tag{3}$$

This is a Gaussian kernel, where the Euclidean distance is substituted by the kernel distance associated with $k_\ell^+$, which yields a valid kernel [16]. The kernel distance can be computed using the kernel trick according to Equation (3).

We proceed by relating the generalized $\ell$-walk node kernel to walk labelings.

**Proposition 3.9.** *For* $\alpha > 0$ *and* $k_V = k_\delta$, *the equality* $\hat{k}_\ell^+(u, v; \alpha) = 1$ *holds if and only if* $u$ *and* $v$ *are* $\ell$-walk indistinguishable, i.e., $\mathrm{rw}_+^{(\ell)}(u) = \mathrm{rw}_+^{(\ell)}(v)$.

*Proof.* For $\alpha \neq 0$, the Gaussian kernel is one if and only if the Euclidean distance is zero, i.e., $\phi_\ell^+(u) = \phi_\ell^+(v)$. With the feature map of $k_\ell^+$ that maps $v$ to the characteristic vector of the multiset $\mathrm{rw}_+^{(\ell)}(v)$ the result follows immediately. $\qquad\square$

Other normalized kernels $\tilde{k}$ satisfying $\tilde{k}(x, y) = 1$ if and only if $x = y$ are known [1, 11], but cannot be controlled with a parameter $\alpha$ in a similar way. For $\alpha = 0$, we have $\hat{k}_\ell^+(u, v) = 1$. For $\alpha \to \infty$, we have $\hat{k}_\ell(u, v) = 0$ unless $u$ and $v$ are $\ell$-walk indistinguishable. In particular, we can relate the generalized $\ell$-walk node kernel to walk labelings.

**Corollary 3.10.** *For* $\alpha \to \infty$, *the image of* $\hat{k}_\ell^+$ *is* $\{0, 1\}$ *and* $\hat{k}_\ell^+(u, v) = 1 \iff \mathrm{rw}_+^{(\ell)}(u) = \mathrm{rw}_+^{(\ell)}(v)$.

### 3.2.1 Computation

Computing walk labelings for labeled graphs by enumerating walks is prohibitive. We employ a technique based on direct product graphs introduced for the computation of random walk kernels, see Section 2.2. Using Lemma 2.3 we can compute the generalized $\ell$-walk node kernel by counting the walks in the direct product graph with the techniques described above. Algorithm 1 realizes this approach and computes the kernel for all pairs of nodes having a non-zero value. The vectors $\boldsymbol{w}$ and $\boldsymbol{w}^+$ each have one entry for each node $(u, v)$ in the direct product graph, which after the $i$th iteration stores the value of $k_i(u, v)$ and $k_i^+(u, v)$, respectively. In Line 8 the generalized $\ell$-walk node kernel is computed according to Definition 3.8 based on the three values $k_i^+(u, v)$, $k_i^+(u, u)$ and $k_i^+(v, v)$ using the kernel trick. The symmetry of kernels is reflected by symmetries in the direct product graph $G \times G$ which can be exploited to speed-up computation. Line 9 is optional and increases expressivity as discussed in Section 3.2.2. The method of computation can be extended to incorporate vertex and edge kernels by using a weighted adjacency matrix and initializing $\boldsymbol{w}^{(0)}$ according to the vertex kernel, see [25] for details. Moreover, the limit $\ell \to \infty$ can be computed by introducing suitable weights and well-known techniques proposed for random walk kernels, see Section 2.2.

### 3.2.2 Weisfeiler-Leman expressiveness

We can modify Algorithm 1 to perform a re-encoding in every iteration to achieve Weisfeiler-Leman expressiveness for $\alpha \to \infty$ by adding Line 9 (highlighted in blue).

**Proposition 3.11.** *For* $\ell > 0$ *and* $\alpha \to \infty$, *Algorithm 1 including Line 9 computes* $\hat{k}_\ell^{+WL}$ *with image* $\{0, 1\}$ *and* $\hat{k}_\ell^{+WL}(u, v) = 1 \iff \mathrm{wl}^{(\ell)}(u) = \mathrm{wl}^{(\ell)}(v)$.

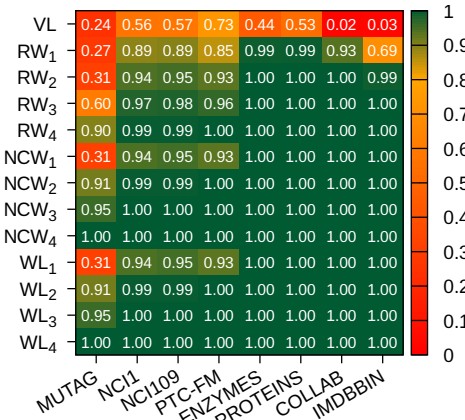

Figure 2: Expressiveness

*Proof.* The construction of the direct product graph and Corollary 3.10 with Proposition 3.7 guarantee that the statement is satisfied initially for $i = 1$. For $\alpha \to \infty$, $\hat{K}_{uv}^{(i)} = 1$ if and only if $w_{uu}^+ = w_{vv}^+ = w_{uv}^+$ in iteration $i$, and zero otherwise. Assume that the equivalence holds in iteration $i$, then the vector $\boldsymbol{w}$ contains ones for the node pairs $(u, v)$ with $\mathsf{wl}^{(i)}(u) = \mathsf{wl}^{(i)}(v)$ and $w_{uu}^+ = w_{vv}^+ = w_{uv}^+$. Hence, in iteration $i + 1$ after Line 4, $\boldsymbol{w}$ counts for each node pair the number of matching neighbors regarding $\mathsf{wl}^{(i)}$. After adding these values to $\boldsymbol{w}^+$, it directly follows that, if $\mathsf{wl}^{(i+1)}(u) = \mathsf{wl}^{(i+1)}(v)$, then $w_{uu}^+ = w_{vv}^+ = w_{uv}^+$ and $\hat{K}_{uv}^{(i+1)} = 1$ in Line 8. Vice versa, $\hat{K}_{uv}^{(i+1)} = 1$ if and only if $w_{uu}^+ = w_{vv}^+ = w_{uv}^+$, which implies $w_{uu} = w_{vv} = w_{uv}$. In combination with $\sigma(u) = \sigma(v)$ from Definition 2.2, we conclude $\mathsf{wl}^{(i+1)}(u) = \mathsf{wl}^{(i+1)}(v)$. $\qquad\square$

## 4 Comparing graphs by walks

We define a graph kernel based on the generalized $\ell$-walk node kernel obtained from the Cartesian product of the node sets of the two input graphs. Its parameters allow controlling the strictness of neighborhood comparison and the importance of walk counts.

**Definition 4.1** (Node-centric $\ell$-walk graph kernel). *Given* $\alpha, \beta \in \mathbb{R}_{\geq 0}$, *the* node-centric $\ell$-walk graph kernel *is defined as*

$$K_\ell^+(G, H; \alpha, \beta) = \sum_{i=0}^{\ell} \sum_{u \in V(G)} \sum_{v \in V(H)} \hat{k}_i^+(u, v; \alpha) k_i^\beta(u, v). \tag{4}$$

The node-centric $\ell$-walk graph kernel combines the generalized $\ell$-walk node kernel of Definition 3.8 with a polynomial of the kernel $k_i$. It can incorporate continuous attributes via a dedicated node kernel $k_V$, cf. Equation (1). We show in the following that Definition 4.1 resembles two widely-used graph kernels from the literature for certain parameter choices.

### 4.1 Relation to random walk and Weisfeiler-Leman subtree kernels

Random walk kernels, see Section 2.2, are closely related to the node-centric $\ell$-walk graph kernel.

**Proposition 4.2.** *The node-centric $\ell$-walk graph kernel for $\alpha = 0$ and $\beta = 1$ is equal to the $\ell$-step random walk kernel with $\lambda_i = 1$ for $i \in \{0, \dots, \ell\}$, i.e., $K_\ell^+(G, H; 0, 1) = K_\times^\ell(G, H)$.*

*Proof.* The parameter choice guarantees that $\hat{k}_i^+(u, v; \alpha) k_i^\beta(u, v) = k_i(u, v)$. Hence, this kernel sums over all pairs of walks $(w, w')$ of length less or equal to $\ell$. Since $k_\delta(\sigma(w), \sigma(w')) = \prod_j k_\delta(\sigma(u_j), \sigma(v_j))$, where $w = (u_0, u_1, \dots)$, $w' = (v_0, v_1, \dots)$, this corresponds to Definition 2.1. $\qquad\square$

This result also holds for the limit $\ell \to \infty$ if we assume that walks are adequately down-weighted by their length to guarantee convergence.

In random walk kernels a pair of vertices, that both have a large neighborhood, have a great ability to contribute strongly to the total kernel value, since a large number of walks originates at them. In the Weisfeiler-Leman subtree kernel the same pair contributes a value bounded by $\ell$ depending on whether their neighborhood matches exactly. We hypothesize that this conceptional difference is crucial and allows the Weisfeiler-Leman kernel to outperform previous walk-based kernels. By setting $\alpha$ sufficiently high and $\beta = 0$, it follows from Corollary 3.10 that the node-centric $\ell$-walk graph kernel resembles the Weisfeiler-Leman kernel in this respect, but uses walk labelings instead. Our discussion of walk labelings and their relation to Weisfeiler-Leman labels suggest that these are only slightly less expressive. We verify our hypothesis experimentally in Section 5. Moreover, by applying the iterated re-encoding described in Section 3.2.2, we obtain exactly the same value as the Weisfeiler-Leman subtree kernel for $\alpha \to \infty$ and $\beta = 0$. Other parameter choices allow for a less strict comparison, which is not possible with the Weisfeiler-Leman subtree kernel.

## 4.2 Computation

We can compute the node-centric $\ell$-walk graph kernel for two graphs $G$ and $H$ by applying Algorithm 1 to their union $G \cup H$ and plugging the results stored in the two kernel matrices $\boldsymbol{K}^{(i)}$ and $\hat{\boldsymbol{K}}^{(i)}$ into Equation (4). This can be optimized by exploiting the structural properties of the direct product graph.

**Proposition 4.3.** *Let $G$ and $H$ be two graphs, then the direct product graph of their union $P = G \cup H$ is $P \times P = (G \times G) \cup (G \times H) \cup (H \times G) \cup (H \times H)$.*

*Proof.* The result follows from Definition 2.2 with the fact that the vertex and edge sets of $G$ and $H$ are disjoint. □

As there are no walks between disconnected components, we can run Algorithm 1 separately on the product graphs $G \times H$, $G \times G$ and $H \times H$, ignoring $H \times G$ due to symmetry. Moreover, if we compute the kernel for all pairs of graphs in a dataset $\mathcal{D}$ of graphs, we can compute all node self-similarities from product graphs $G \times G$ for all $G \in \mathcal{D}$ once in a preprocessing step. In this case, the overhead compared to the standard $\ell$-step random walk kernel is only minor and is mainly attributed to exponentiation. Direct computation via the directed product graph is not suitable for large-scale graphs [25]. However, existing acceleration techniques [43, 20, 19] remain applicable. In particular, for unlabeled graphs (or $k_V(u, v) = 1$ for all nodes $u, v$) the result can be obtained from the walk counts of the input graphs without generating a product graph.

# 5 Experimental evaluation

We experimentally verify the hypotheses that have originated from our theoretical results. In particular, we aim to answer the following research questions.

**Q1** Is the Weisfeiler-Leman subtree kernel more expressive than walk-based kernels regarding their ability to distinguish non-isomorphic graphs on common benchmark datasets?

**Q2** Are our walk-based kernels competitive to the state-of-the-art regarding classification accuracy?

**Q3** Which level of strictness in walk-based neighborhood comparison is most suitable?

## 5.1 Experimental setup

We describe the kernels under comparison, their parameters and the used datasets. All experiments were performed on an Intel Xeon E5-2690v4 machine at 2.6GHz with 384 GB of RAM.

### 5.1.1 Kernels

We implemented the node-centric $\ell$-walk kernel (NCW) of Definition 4.1 with $\hat{k}_i^+$ and $k_i$ computed by Algorithm 1 without the blue Line 9 and the variant with WL expressiveness including the blue line (NCWWL). For unlabeled graphs we computed NCW without creating the direct product graph.

Table 1: Average classification accuracies in percent and standard deviations. The highest accuracy of each dataset is marked in **bold**; OOM denotes an out of memory error.

| METHOD | DATASETS | | | | | | | |
|---|---|---|---|---|---|---|---|---|
| | MUTAG | NCI1 | NCI109 | PTC-FM | ENZYMES | PROTEINS | IMDBBIN | COLLAB |
| VL | $85.4\pm0.7$ | $64.8\pm0.1$ | $63.6\pm0.2$ | $58.0\pm0.7$ | $23.6\pm0.9$ | $72.1\pm0.2$ | $46.7\pm1.1$ | $56.2\pm0.0$ |
| EL | $85.5\pm0.6$ | $66.4\pm0.1$ | $64.9\pm0.1$ | $57.8\pm0.7$ | $27.7\pm0.7$ | $73.5\pm0.3$ | $45.9\pm0.8$ | $61.7\pm0.2$ |
| SP | $83.3\pm1.4$ | $74.3\pm0.3$ | $73.3\pm0.1$ | $60.4\pm1.7$ | $39.2\pm1.4$ | $73.8\pm0.4$ | $48.0\pm0.8$ | $68.2\pm0.1$ |
| GH | $85.4\pm1.9$ | $72.8\pm0.2$ | $71.7\pm0.3$ | $57.8\pm1.2$ | $33.9\pm1.0$ | $68.1\pm0.5$ | $52.6\pm0.8$ | $65.9\pm0.4$ |
| GL3 | $85.2\pm0.9$ | $70.5\pm0.2$ | $69.3\pm0.2$ | $57.9\pm1.4$ | $30.1\pm1.2$ | $72.9\pm0.4$ | $50.7\pm0.8$ | $66.7\pm0.1$ |
| WL | $87.0\pm1.9$ | $85.3\pm0.3$ | $85.9\pm0.3$ | $\mathbf{63.7}\pm1.3$ | $53.8\pm1.2$ | $\mathbf{75.1}\pm0.3$ | $\mathbf{71.6}\pm0.8$ | $79.0\pm0.1$ |
| RW | $\mathbf{87.8}\pm0.9$ | $75.4\pm0.2$ | $74.5\pm0.1$ | $57.3\pm1.7$ | $33.9\pm0.9$ | $73.4\pm0.6$ | $68.4\pm0.5$ | $56.2\pm0.0$ |
| NCW | $86.9\pm0.9$ | $\mathbf{85.5}\pm0.2$ | $85.9\pm0.2$ | $63.4\pm1.2$ | $54.8\pm1.0$ | $74.8\pm0.5$ | $70.4\pm0.8$ | $\mathbf{79.4}\pm0.1$ |
| NCWWL | $87.1\pm1.2$ | $\mathbf{85.5}\pm0.2$ | $\mathbf{86.3}\pm0.2$ | $62.3\pm1.0$ | $\mathbf{55.2}\pm1.2$ | $74.8\pm0.5$ | $\mathbf{71.6}\pm0.6$ | OOM |

As a baseline we used the node label kernel (VL) and edge label kernel (EL), which are the dot products on node and edge label histograms, respectively, see [41, 26]. For the Weisfeiler-Leman subtree kernel (WL), the $\ell$-step random walk kernel (RW) as well as NCW and NCWWL we chose the iteration number and walk length from $\{0, \ldots, 5\}$ by cross-validation. For RW, $\lambda_i = 1$ for $i \in \{0, \ldots, \ell\}$ was used. For NCW and NCWWL, we selected $\alpha$ from $\{0.01, 0.1, 1, 1000\}$ and $\beta$ from $\{0, 0.5, 1\}$. We have not included extensions of the WL such as [24, 42], which could also be applied similarly to the node-centric $\ell$-walk graph kernel. In addition we used a graphlet kernel (GL3), the shortest-path kernel (SP) [4], and the Graph Hopper kernel (GH) [9]. GL3 is based on connected subgraphs with three nodes taking labels into account similar to the approach used by Shervashidze et al. [40]. For SP and GH we used the Dirac kernel to compare path lengths and node labels. We implemented the node-centric $\ell$-walk graph kernel as well as all baselines in Java.[1] We performed classification experiments with the $C$-SVM implementation LIBSVM [6]. We report mean prediction accuracies and standard deviations obtained by 10-fold nested cross-validation repeated 10 times with random fold assignment. Within each fold all necessary parameters were selected by cross-validation based on the training set. This includes the regularization parameter $C$ and kernel parameters.

### 5.1.2 Datasets

We tested on widely-used graph classification benchmarks datasets of the TUDATASETS repository [32] representing graphs from different domains. MUTAG, NCI1, NCI109 and PTC-FM represent small molecules, ENZYMES and PROTEINS are derived from macromolecules, and COLLAB and IMDBBIN are social networks. The datasets define binary graph classification experiments with exception of ENZYMES and COLLAB, which are divided into six and three classes, respectively. All graphs have node labels with exception of the social network graphs. We removed edge labels, if present, since they are not supported by all graph kernel implementations.

### 5.2 Results

We discuss our results and research questions.

**Q1: Expressiveness.** We investigate the expressiveness of the $\ell$-step random walk kernel ($\text{RW}_\ell$), the node-centric $\ell$-walk graph kernel ($\text{NCW}_\ell$) with $\alpha = 1000$ and $\beta = 0$, and the Weisfeiler-Leman subtree kernel ($\text{WL}_\ell$). Since the used graph datasets contain duplicates [18], we filtered them such that for each isomorphism class a single representative remains. We computed the *completeness ratio* introduced by Kriege et al. [26], i.e., the fraction of graphs that can be distinguished from all other graphs in the dataset. Figure 2 confirms that with increasing parameter $\ell$ all methods become more expressive. For all datasets we observe that $\text{NCW}_\ell$ is clearly more expressive than $\text{RW}_\ell$ showing that grouping the walks according to their start node increases the expressive power. Moreover, $\text{NCW}_\ell$ achieves the same values as $\text{WL}_\ell$ for all $\ell$. Although we have shown that walk labelings are weaker than Weisfeiler-Leman labelings, cf. Figure 1, this does not affect the ability to distinguish the considered real-world graphs. This confirms that walks provide expressive features when grouped at a node level.

---

[1]Our code is publicly available at `https://github.com/nlskrg/node_centric_walk_kernels`.

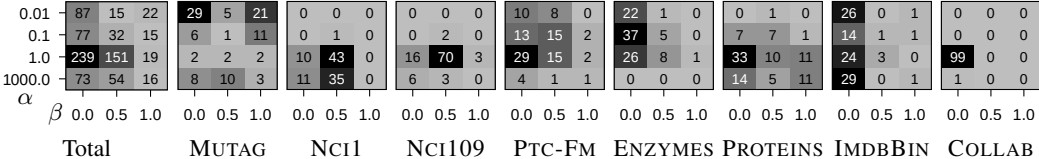

Figure 3: Selection of the parameters $\alpha$ (y-axis) and $\beta$ (x-axis) of the node-centric $\ell$-walk graph kernel on different datasets. Note that the combination at the bottom left resembles Weisfeiler-Leman type kernels and the combination at the top right the $\ell$-step random walk kernel.

**Q2: Accuracy.** Table 1 shows the resulting accuracies and standard deviations. For several datasets WL outperforms RW as expected, e.g., NCI1, NCI109 and ENZYMES. In these cases NCW provides an accuracy close to WL and outperforms RW by a large margin, although both are based on walks and NCW is obtained with only a minor modification of RW. This clearly shows that node-centric walks are suitable for obtaining high accuracies. For ENZYMES, NCW reaches an accuracy higher than WL indicating that a higher expressiveness does not necessarily lead to better generalization. This suggests that a less strict comparison of neighborhoods can be beneficial for some of the datasets. NCWWL overall performs comparable to NCW but reaches slight improvements for several datasets.

**Q3: Strict neighborhood comparison.** NCW supports non-strict neighborhood comparison by tuning the parameter $\alpha$ and incorporates walk counts for increasing $\beta$ leading to a kernel similar to WL and the $\ell$-step random walk kernel for the extreme cases. We investigate the selection of these parameters in the above classification experiments. For each dataset 10-fold cross-validation was repeated 10 times leading to 100 classification experiments in total, for each of which the hyperparameters were optimized. The distribution of the selected values for $\alpha$ and $\beta$ is shown in Figure 3. We observe that the choices vary between datasets. For the social network datasets there is a clear preference for $\beta = 0$, which explains the worse performance of the $\ell$-step random walk kernels on these datasets. For NCI1 and NCI109, $\alpha \geq 1$ and $\beta \leq 0.5$ were preferred leading to kernel values closer to WL, which is in accordance with the accuracy reached. Of particular interest is the dataset ENZYMES for which NCW and NCWWL provide the highest accuracies. Figure 3 shows that for this dataset, in most cases non-strict neighborhood comparison ($\alpha \leq 1$) with only a minor influence of walk counts ($\beta \leq 0.5$) was selected, a choice not possible with previous kernels. Further investigation shows that the parameter $\ell = 2$ was selected in the majority of the cases for WL, while for NCW, $\ell = 4$ with $\alpha = 0.1$ or $\alpha = 0.01$ was selected most frequently. This indicates that for this dataset it is preferable to take larger neighborhoods into account but to compare them less strictly.

## 6 Conclusion and future work

We have contributed to the understanding of walk and Weisfeiler-Leman based methods showing that classical random walk kernels can be lifted to obtain the expressive power of the Weifeiler-Leman subtree kernel, drastically increasing classification accuracy. While the direct product graph based approach is less efficient than the Weisfeiler-Leman method, its advantage is that we can control the strictness of neighborhood comparison and incorporate node and edge similarities for attributed graphs. Improving the running time and the application to attributed graphs remains future work.

Our results have implications beyond graphs kernels. In graph neural networks, the *bottleneck problem* [2] refers to the observation that node neighborhoods grow exponentially with increasing radius and thus cannot be represented accurately by a fixed-sized feature vector. This is a possible explanation for the weakness of GNNs in long-range tasks. The product graph based approach works with infinite-dimensional node embeddings using the kernel trick. Investigating the bottleneck problem in product graph based GNNs might shed more light into this phenomenon.

## Acknowledgements

I would like to thank Floris Geerts for pointing me to the paper on walk partitions by Powers and Sulaiman [36]. I am grateful to the Vienna Science and Technology Fund (WWTF) for supporting me through project VRG19-009.

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
