# OpenReview forum: "Weisfeiler and Leman Go Walking: Random Walk Kernels Revisited"
_NeurIPS.cc/2022/Conference — NeurIPS 2022 Accept_

### Official Review · Reviewer_58fn · 2022-07-10

**Rating:** 7
**Confidence:** 3
**Soundness:** 3 good
**Presentation:** 3 good
**Contribution:** 3 good

**Summary:**

This paper gives a unified view on random walk kernels and Weisfeiler-Leman subtree kernel and relate walk-based label refinement to the Weisfeiler-Leman method. They derive a node-centric walk kernel on graphs which generalizes the classical random walk kernel and allows grouping of walks by their start node. They verify the theoretical results on real-world graphs and show that their method has higher accuracies, surpassing the Weisfeiler-Leman subtree kernel in some cases.

**Questions:**

We do not have questions.

**Limitations:**

They address it adequately.

**Strengths And Weaknesses:**

Originality and Significance: This work is the first that relate random walks in labeled graphs and the Weisfeiler-Leman method formally and show that classical random walk kernels with only minor modifications regarding definition and computation are as expressive as the widely-used Weisfeiler-Leman subtree kernel but support non-strict neighborhood comparison. The contributions play well into previous work as a helpful summary as well as extension and also bring inspiration to future research.
Quality: The paper is well done with sufficiently proved theoretical results and the detailed experiments which verified the key problems the paper concerned(Q1-Q3).
Clarity: Clear writing and structure.

**Weaknesses:** Some notations are recommended to be mentioned like $  \Delta $ in line 109.

---

### Official Review · Reviewer_cwFz · 2022-07-11

**Rating:** 9
**Confidence:** 5
**Soundness:** 4 excellent
**Presentation:** 4 excellent
**Contribution:** 4 excellent

**Summary:**



The paper formally relates finite length node centered random walks to the Weisfeiler Lehman method.
The Weisfeiler Lehman vertex relabeling algorithm produces a hierarchy of vertex labels, which is shown to be related to a vertex label hierarchy that can be obtained from multisets of vertex centered walks, which correspond to root-leaf paths in the unfolding trees of various depths.
The paper subsequently develops variants of a vertex kernel that is based on multisets of labeled walks, relates them to the random walk, as well as the weisfeiler lehman subtree kernel, and empirically shows that it performs well in practice.

**Questions:**

Proposition 3.9 is correct if the base kernel used for vertex comparison within the walks is $k_V$ as defined in Line 202. However, it seems that using a different base kernel, such as $k(u,v) = 0 \ \forall u,v \in V(G)$ should yield $\hat{k}^+_\ell(u,v,a) = 1 \  \forall u,v \in V(G) \ \forall \alpha\geq 0$. Therefore, for this (obviously useless) base kernel, Proposition 3.9 would not hold. It hence seems that Line 202 'w.l.o.g' might be misleading. Furthermore, it should be stressed that Prop. 3.9 is somewhat specific to the base kernel $k_V$ (as it is also used in the proof).
In fact, any kernel that identifies two or more vertex labels will result similar issues.
Is my assessment correct?


In the experimental section, I am a bit lost. I am assuming that NCW is the kernel as of Definition 4.1 with $\hat{k}^+_i$ and $k_i$ computed by Algorithm 1 _without the blue Line 9_, while NCWWL is _the same, with Line 9_?
It might reduce (at least my) potential confusion if you could clearify this a bit more.

In the experimental evaluation of the expressivity, it seems that $\alpha = 1000$ is a rather large parameter for the rbf part of Def 4.1. It seems to be the largest parameter that you consider in the experiments. Why do you choose $\alpha=1000$ and $\beta=0$ here? How does the expressiveness change if the parameters are changed? Also, how does NCWWL behave?


The paper is very nicely and cleanly written. It has been a real pleasure reading it! Minor typos and issues that I have found are listed below:

- l58: Our Contribution -> Our contribution
- l143/Fig.1: It seems that Fig(1a) shows the unfolding trees of nodes $f$ and $e$, not $f$ and $g$, as mentioned in line 143. The text is correct for $f$ and $g$. I was confused for a while.
- l223: a techniques
- l228: line -> Line
- l300: TuDatsets (also in appendix somewhere)
- Fig2: the Label kernel is not described in the experimental section. Is it VL, EL, or a combination of the two?


**Limitations:**


The authors don't report runtimes of the kernel computation and mention runtime improvements as future work. Another mentioned limitation is the current restriction to node label information and to discretely labeled graphs. It seems that these limitations are not inherent to the node centric $\ell$-walk graph kernel, but merely due to implementation and/or scope of the experiments.


**Strengths And Weaknesses:**

The paper is very well written and technically very precise. It was a pleasure to read and easy to follow.
Every step is very clearly presented and seems to follow naturally. However, in the almost twenty (resp. fifteen) years since the introduction of random walk (resp. WL subtree) kernels, nobody (as of my knowledge) has yet drawn this connection this rigorously. This insight deserves to be published at a very good conference.

A generalized $\ell$-walk node kernel is proposed as a result of the insights on the connection between WL and RW refinement hierarchies.
It can be seen as a soft (a little bit weaker) version of the discrete WL vertex kernel. It is hence related to recent approaches which try to weaken the strict dirac-delta comparison of node labels/unfolding trees of the WL subtree kernel. The resulting node centric $\ell$-walk graph kernel is then shown to generalize the random walk kernel, as well as the Weisfeiler Lehman subtree kernel in the limit $\alpha \to \infty$.
As a result of the comprehensive view that the authors have, it can be readily computed using the machinery that was developed over the last decades.
Empirical evaluation suggests that the theoretical results on expressiveness translate to comparable predictive performance and expressiveness in practice.

The only 'weakness' that I want to briefly mention is that now that this connection is made, there is an open field of practical questions that could be addressed, regarding the application of 'Weisfeiler-Lehman-like' kernels to attributed graphs, which can now potentially be addressed by choice of suitable base kernels.
While certainly out of scope of this article, I am eager to see more.

---

> ### Author Response · Authors · 2022-08-02
> **Response to questions**
>
> Thank you very much for your constructive and encouraging review!
>
> ### Observation regarding Proposition 3.9
>
> Your assessment is perfectly correct. Thank you for pointing out this flaw. We will include the assumption $k_V=k_\delta$ in the proposition.
>
> ### Experimental evaluation
> > In the experimental section, I am a bit lost. I am assuming that NCW is the kernel as of Definition 4.1 with $\hat{k}^+_i$
> and $k_i$ computed by Algorithm 1 *without the blue Line 9*, while NCWWL is *the same, with Line 9*?
>
> Yes, this is correct. We will clarify this in a revised version of the paper.
>
> > In the experimental evaluation of the expressivity, it seems that $\alpha=1000$ is a rather large parameter for the rbf part of Def 4.1. It seems to be the largest parameter that you consider in the experiments. Why do you choose $\alpha=1000$ and $\beta=0$ here? How does the expressiveness change if the parameters are changed? Also, how does NCWWL behave?
>
> By setting $\alpha$ sufficiently high and $\beta=0$, NCW becomes similar to the WL subtree kernel, with the only difference that walk labels are used instead of WL labels. This motivated the choice of the parameters, and the experiments show that both approaches are equally expressive on the used real-world datasets. However, the *same expressiveness* is reached for all $\alpha>0$, since the Gaussian RBF kernel of Def. 3.8 is universal in this case (the choice of $\beta$ is irrelevant). We were able to verify this experimentally when taking numerical inaccuracies for some parameter combinations into account. NCWWL shows the same behavior.
>
>
> > Fig2: the Label kernel is not described in the experimental section. Is it VL, EL, or a combination of the two?
>
> The Label kernel is VL (corresponding to RW$_0$, NCW$_0$ and WL$_0$); we will clarify this in a revised version and also fix the other minor issues.

---

### Official Review · Reviewer_rmAL · 2022-07-11

**Rating:** 5
**Confidence:** 4
**Soundness:** 3 good
**Presentation:** 3 good
**Contribution:** 2 fair

**Summary:**

The paper presents a novel framework to describe random walk kernels. First, a unified view is provided to integrate and relate random walk kernels with the popular Weisfeler-Lehman (WL) framework. The paper also introduces a modification of the classical random walk kernel, which allows to improve the expressivity to match the performance of the WL kernel. Experimental analysis is showed to compare the proposed framework with existing graph kernels.

**Questions:**

Have the authors compared to the GraphHopper kernel? The kernel formulation seem quite similar, in terms of adding a label-based similarity. I guess one can see the current kernel as an extension of the RW kernel, as the GraphHopper kernel is an extension of the classical shortest path.

I believe that a stronger experimental section might help to show the actual benefit of the approach. The impact of the paper in the current status is moderated. In particular, I would suggest:

- Extensive runtime evaluation: this is especially important given that he RW kernels are computationally expensive
- Application on attributed graphs: given that the performance superiority to WL is limited, this additional use case would be beneficial  to improve the experimental comparison and show the benefit of the approach.
- Potentially, find simulated or real case graphs where the kernel can capture information missed by WL

**Limitations:**

No negative societal impact.

**Strengths And Weaknesses:**

Strengths:

- The paper is well written and properly structured, with good notation and clarity of presentation
- Both the experimental evaluation and theoretical framework are solid and robust

Weaknesses:

- The originality is limited. While the integration of these kernel approaches in a unified view is innovative, the methodological changes in the kernel are incremental
- The experimental results are also limited. The actual benefit of the modified RW kernel is not reflected in the results.

---

> ### Author Response · Authors · 2022-08-02
> **Response to weaknesses and questions**
>
> > The originality is limited. While the integration of these kernel approaches in a unified view is innovative, the methodological changes in the kernel are incremental
>
> We have intentionally kept close to the method of RW kernels to demonstrate that small (or incremental) changes lead to a significant improvement in expressiveness and accuracy.
>
>
> > Have the authors compared to the GraphHopper kernel? The kernel formulation seem quite similar, in terms of adding a label-based similarity. I guess one can see the current kernel as an extension of the RW kernel, as the GraphHopper kernel is an extension of the classical shortest path.
>
> Several papers have used label-based similarities with RW kernels, e.g. [14, 40, 22]. Our kernel groups walks at their starting nodes, increasing expressiveness and accuracy over standard formulations. This grouping technique is neither used by the classical shortest-path kernel nor by the GraphHopper kernel but could also be useful for these kernels. Please find below the results of the GraphHopper kernel, which performs clearly worse than our approach (NCW) on the considered datasets.
>
> |              | Mutag        | Nci1         | Nci109       | Ptc-Fm       | Enzymes      | Proteins     | ImdbBin      |
> |--------------|--------------|--------------|--------------|--------------|--------------|--------------|--------------|
> | GraphHopper  | 85.4$\pm$1.9 | 72.8$\pm$0.2 | 71.7$\pm$0.3 | 57.8$\pm$1.2 | 33.9$\pm$1.0 | 68.1$\pm$0.5 | 52.6$\pm$0.8 |
> |  NCW         | 86.9$\pm$0.9 | 85.5$\pm$0.2 | 85.9$\pm$0.2 | 63.4$\pm$1.2 | 54.8$\pm$1.0 | 74.8$\pm$0.5 | 70.4$\pm$0.8 |
>
> > Extensive runtime evaluation: this is especially important given that he RW kernels are computationally expensive
>
> We agree that walk-based kernels do not scale to very large graphs. Several works address this issue to a certain extent and remain applicable also for our kernels. The overhead of our modification using some optimizations briefly discussed in Section 4.2 is small.
>
> > Application on attributed graphs: given that the performance superiority to WL is limited, this additional use case would be beneficial to improve the experimental comparison and show the benefit of the approach.
>
> We agree that the application to attributed graphs is particularly promising. Due to space and time limitations, we will not be able to provide an extensive comparison on attributed graphs and plan that as future work.

---

### Official Review · Reviewer_kGem · 2022-07-14

**Rating:** 5
**Confidence:** 3
**Soundness:** 2 fair
**Presentation:** 3 good
**Contribution:** 1 poor

**Summary:**

The paper proposes a walk-based kernels for graphs with nodes being labeled with WL labeling scheme. It is supposed to improve expressiveness of random walk kernels and can be as good as WL subtree kernels.

--------------------
After looking at responses, other reviews and the paper again, I do find the paper does offer a new way to compute kernels, even though not strictly solving the expressiveness problem of previous work, may offer a new view and future development. I think I can raise my evaluation a bit.

**Questions:**

What is the benefit of the proposed kernels? What does controlling the strictness of neighborhood comparison help classifying graphs in practice? What's the benefit of node/edge similarities?


**Ethics Review Area:**

["I don’t know"]

**Limitations:**

The paper is clear about its limitations.

**Strengths And Weaknesses:**

The paper's originality is in its combination of walk-based kernels with WL labeling scheme. This is proved to be an improvement on random walk kernels, but not on WL-based kernels.

The paper show many properties of the proposed methods, which I could not check in details, can be interesting. The paper's writing is clear.

However, all of these do not seem to solve any problem of graph kernels. The approach is shown that can be as good as WL kernels with more computation. I cannot see the benefit of the paper in practice.

---

> ### Author Response · Authors · 2022-08-02
> **Clarification and response to questions**
>
> > The paper proposes a walk-based kernels for graphs with nodes being labeled with WL labeling scheme.
> > The paper's originality is in its combination of walk-based kernels with WL labeling scheme.
>
> To clarify a possible misunderstanding, we would like to emphasize that we do not use WL labels in our random walk kernel but show that walks themselves already capture (almost) all the information contained in WL labels (given that walk counts are normalized and summed up correctly).
>
> > What is the benefit of the proposed kernels? What does controlling the strictness of neighborhood comparison help classifying graphs in practice? What's the benefit of node/edge similarities?
>
> The general benefit of supporting node/edge similarities has been shown in several papers focusing on attributed graphs, e.g., (Feragen et al., Scalable kernels for graphs with continuous attributes. NIPS 2013). Composing kernels for complex objects from kernels on their parts, including their attributes, is supported by very few graph kernels and the classical random walk kernel is one of them. Our results for the ENZYMES dataset indicate that there are classes of graphs for which non-strict neighborhood comparison is beneficial. This is supported by the improvement in accuracy and the selection of the hyperparameters $\alpha$ and $\beta$.

---

### Author Response · Authors · 2022-08-02
**Thank you for the constructive reviews**

We would like to thank all reviewers for their comments and helpful feedback on our manuscript. Individual points are addressed below in response to the reviews.

---

### Meta-Review · Area_Chair_i5de · 2022-08-28

**Recommendation:** Accept
**Confidence:** Certain

**Metareview:**

 The paper yields new understanding of random walk kernels and W-L graph isomorphism test based kernels. This is a very interesting contribution towards understanding kernels between graphs. This should be of interest to Neurips community.

**Award:**

No

---

### Decision · Program_Chairs · 2022-09-14

Accept